# How Many Samples is a Good Initial Point Worth in Low-rank Matrix Recovery?

**Gavin Zhang**
Department of Electrical and Computer Engineering
University of Illinois Urbana Champaign
Illinois, IL61820
jialun2@illinois.edu

**Richard Y. Zhang**
Department of Electrical and Computer Engineering
University of Illinois Urbana Champaign
Illinois, IL61820
ryz@illinois.edu

## Abstract

Given a sufficiently large amount of labeled data, the non-convex low-rank matrix recovery problem contains no spurious local minima, so a local optimization algorithm is guaranteed to converge to a global minimum starting from any initial guess. However, the actual amount of data needed by this theoretical guarantee is very pessimistic, as it must prevent spurious local minima from existing anywhere, including at adversarial locations. In contrast, prior work based on good initial guesses have more realistic data requirements, because they allow spurious local minima to exist outside of a neighborhood of the solution. In this paper, we quantify the relationship between the quality of the initial guess and the corresponding reduction in data requirements. Using the restricted isometry constant as a surrogate for sample complexity, we compute a sharp "threshold" number of samples needed to prevent each specific point on the optimization landscape from becoming a spurious local minimum. Optimizing the threshold over regions of the landscape, we see that for initial points around the ground truth, a linear improvement in the quality of the initial guess amounts to a constant factor improvement in the sample complexity.

## 1 Introduction

A perennial challenge in non-convex optimization is the possible existence of *bad* or *spurious* critical points and local minima, which can cause a local optimization algorithm like gradient descent to slow down or get stuck. Several recent lines of work showed that the effects of non-convexity can be tamed through a large amount of diverse and high quality training data [17, 1, 9, 3, 18, 12]. Concretely, these authors showed that, for classes of problems based on random sampling, spurious critical points and local minima become progressively less likely to exist with the addition of each new sample. After a *sufficiently large number of samples*, all spurious local minima are eliminated, so any local optimization algorithm is guaranteed to converge to the globally optimal solution starting from an arbitrary, possibly random initial guess.

This notion of a *global* guarantee—one that is valid starting from any initial point—is considerably stronger than what is needed for empirical success to be observed [8]. For example, the existence of a spurious local minimum may not pose an issue if gradient descent does not converge towards it.

However, a theoretical guarantee is no longer possible, as starting the algorithm from the spurious local minimum would result in failure [22]. As a consequence, these global guarantees tend to be pessimistic, because the number of samples must be sufficiently large to eliminate spurious local minima everywhere, even at adversarial locations. By contrast, the weaker notion of a *local* guarantee [11, 10, 15, 19, 5, 7, 20, 13]—one that is valid only for a specified set of initial points—is naturally less conservative, as it allows spurious local minima to exist outside of the specified set.

In this paper, we provide a unifying view between the notions of the global and local guarantees by quantifying the relationship between the sample complexity and the quality of the initial point. We restrict our attention to the *matrix sensing* problem, which seeks to recover a rank-$r$ positive semidefinite matrix $M^* = ZZ^T \in \mathbb{R}^{n \times n}$ with $Z \in \mathbb{R}^{n \times r}$ from $m$ sub-Gaussian linear measurements of the form

$$b \equiv \mathcal{A}(ZZ^T) \equiv [\langle A_1, M^* \rangle \quad \cdots \quad \langle A_m, M^* \rangle]^T \tag{1}$$

by solving the following non-convex optimization problem:

$$\min_{X \in \mathbb{R}^{n \times r}} f_{\mathcal{A}}(X) \equiv \left\| \mathcal{A} \left( XX^T - ZZ^T \right) \right\|^2 = \sum_{i=1}^{m} \left( \langle A_i, XX^T \rangle - b_i \right)^2. \tag{2}$$

We characterize a sharp "threshold" on the number of samples $m$ needed to prevent each specific point on the optimization landscape from becoming a spurious local minimum. While the threshold is difficult to solve, we derive a lower-bound in closed-form based on spurious *critical points*, and show that it constitutes a *sharp* lower-bound on the original threshold of interest. The lower-bound reveals a simple geometric relationship: a point $X$ is more likely to be a local minimum if the column spaces of $X$ and $Z$ are close to orthogonal. Optimizing the closed-form lower-bound over regions of the landscape, we show that for initial points close to the ground truth, a constant factor improvement of the initial point amounts to a constant factor reduction in the number of samples needed to guarantee recovery.

## 2   Related Work

**Local Guarantees**. The earliest work on exact guarantees for non-convex optimization focused on generating a good initial guess within a local region of attraction. For instance, in [21, 24], the authors showed that when $\mathcal{A}$ satisfies $(\delta, 6r)$-RIP with a constant $\delta \leq 1/10$, and there exists a initial point sufficiently close to the ground truth, then gradient descent starting from this initial point has a linear convergence rate. The typical strategy to find such the initial point is *spectral initialization* [11, 10, 21, 19, 5, 14, 6]: using the singular value decomposition on a surrogate matrix to find low-rank factors that are close to the ground truth.

In this paper, we focus on the trade-off between the quality of an initial point and the number of samples needed to prevent the existence of spurious local minima, while sidestepping the question of how it is found. We note, however, that the number of samples needed to find an $\epsilon$-good initial guess (e.g. via spectral initialization) forms an interesting secondary trade-off. It remains a future work to study the interactions between these two points.

**Global Guarantees**. Recent work focused on establishing a global guarantee that is independent of the initial guess [17, 1, 9, 3, 18, 12]. For our purposes, Bhojanapalli et al. [2] showed that RIP with $\delta_{2r} < 1/5$ eliminates all spurious local minima, while Zhang et al. [23] refined this to $\delta_{2r} < 1/2$ for the rank-1 case, and showed that this is both and necessary and sufficient. This paper is inspired by proof techniques in the latter paper; an important contribution of our paper is generalizing their rank-1 techniques to accommodate for matrices of arbitrary rank.

## 3   Our Approach: Threshold RIP Constant

Previous work that studied the global optimization landscape of problem (2) typically relied on the restricted isometry property (RIP) of $\mathcal{A}$. It is now well-known that if the measurement operator $\mathcal{A}$ satisfies the restricted isometry property with a sufficiently small constant $\delta < 1/5$ then problem (2) contains no spurious local minima; see Bhojanapalli et al. [2].

**Definition 1** ($\delta$-RIP). Let $\mathcal{A} : \mathbb{R}^{n \times n} \to \mathbb{R}^m$ be a linear measurement operator. We say that $\mathcal{A}$ satisfies the $\delta$-*restricted isometry property* (or simply $\delta$-RIP) if satisfies the following inequality

$$(1 - \delta)\|M\|_F^2 \leq \|\mathcal{A}(M)\|^2 \leq (1 + \delta)\|M\|_F^2 \qquad \forall M \in \mathcal{M}_{2r}$$

where $\mathcal{M}_{2r} = \{X \in \mathbb{R}^{n \times n} : \text{rank}(X) \leq 2r\}$ denotes the set of rank-$2r$ matrices. The RIP constant of $\mathcal{A}$ is the smallest value of $\delta$ such that the inequality above holds.

Let $\delta \in [0, 1)$ denote the RIP constant of $\mathcal{A}$. It is helpful to view $\delta$ as a surrogate for the number of measurements $m \geq 0$, with a large value of $\delta$ corresponding a smaller value of $m$ and vice versa. For a wide range of sub-Gaussian measurement ensembles, if $m \geq C_0 nr/\delta^2$ where $C_0$ is an absolute constant, then $\mathcal{A}$ satisfies $\delta$-RIP with high probability [4, 16].

Take $X \in \mathbb{R}^{n \times r}$ to be a *spurious* point such that $XX^T \neq ZZ^T$. Our approach in this paper is to define a *threshold* number of measurements that would be needed to prevent $X$ from becoming a local minimum for problem (1). Viewing the RIP constant $\delta$ as a surrogate for the number of measurements $m$, we follow a construction of Zhang et al. [23], and instead define a threshold $\delta_{\text{soc}}(X)$ on the RIP constant $\delta$ that would prevent $X$ from becoming a local minimum for problem (1). Such a construction must necessarily take into account all choices of $\mathcal{A}$ satisfying $\delta$-RIP, including those that adversarially target $X$, bending the optimization landscape into forming a region of convergence around the point. On the other hand, such adversarial choices of $\mathcal{A}$ must necessarily be defeated for a sufficiently small threshold on $\delta$, as we already know that spurious local minima cannot exist for $\delta < 1/5$. The statement below makes this idea precise, and also extends it to a set of spurious points.

**Definition 2** (Threshold for second-order condition). Fix $Z \in \mathbb{R}^{n \times r}$. For $X \in \mathbb{R}^{n \times r}$, if $XX^T = ZZ^T$, then define $\delta_{\text{soc}}(X) = 1$. Otherwise, if $XX^T \neq ZZ^T$, then define

$$\delta_{\text{soc}}(X) \equiv \min_{\mathcal{A}}\{\delta : \nabla f_{\mathcal{A}}(X) = 0, \quad \nabla^2 f_{\mathcal{A}}(X) \succeq 0, \quad \mathcal{A} \text{ satisfies } \delta\text{-RIP}\} \qquad (3)$$

where the minimum is taken over all linear measurements $\mathcal{A} : \mathbb{R}^{n \times n} \to \mathbb{R}^m$. For $\mathcal{W} \subseteq \mathbb{R}^{n \times r}$, define $\delta_{\text{soc}}(\mathcal{W}) = \inf_{X \in \mathcal{W}} \delta_{\text{soc}}(X)$.

If $\delta < \delta_{\text{soc}}(X)$, then $X$ cannot be a spurious local minimum by construction, or it would contradict the definition of $\delta_{\text{soc}}(X)$ as the minimum value. By the same logic, if $\delta < \delta_{\text{soc}}(\mathcal{W})$, then no choice of $X \in \mathcal{W}$ can be a spurious local minimum. In particular, it follows that $\delta_{\text{soc}}(\mathbb{R}^{n \times r})$ is the usual *global* RIP threshold: if $\mathcal{A}$ satisfies $\delta$-RIP with $\delta < \delta_{\text{soc}}(\mathbb{R}^{n \times r})$, then $f_{\mathcal{A}}(X)$ is guaranteed to admit no spurious local minima. Starting a local optimization algorithm from any initial point guarantees exact recovery of an $X$ satisfying $XX^T = ZZ^T$.

Now, suppose we are given an initial point $X_0$. It is natural to measure the *quality* of $X_0$ by its relative error, as in $\varepsilon = \|XX^T - ZZ^T\|_F / \|ZZ^T\|_F$. If we define an $\varepsilon$-neighborhood of all points with the same relative error

$$\mathcal{B}_{\varepsilon} = \{X \in \mathbb{R}^{n \times r}, \|XX^T - ZZ^T\|_F \leq \varepsilon \|ZZ^T\|_F\} \qquad (4)$$

then it follows that $\delta_{\text{soc}}(\mathcal{B}_{\varepsilon})$ is an analogous *local* RIP threshold: if $\mathcal{A}$ satisfies $\delta$-RIP with $\delta < \delta_{\text{soc}}(\mathcal{B}_{\varepsilon})$, then $f_{\mathcal{A}}(X)$ is guaranteed to admit no spurious local minima over all $X \in \mathcal{B}_{\varepsilon}$. Starting a local optimization algorithm from the initial point $X_0$ guarantees either exact recovery of an $X$ satisfying $XX^T = ZZ^T$, or termination at a strictly worse point $X$ with $\|XX^T - ZZ^T\|_F > \|X_0 X_0^T - ZZ^T\|_F$. Imposing further restrictions on the algorithm prevents the latter scenario from occurring (local strong convexity with gradient descent [19], strict decrements in the levels set [10, 23, 8]), and so exact recovery is guaranteed.

The numerical difference between the global threshold $\delta_{\text{soc}}(\mathbb{R}^{n \times r})$ and the local threshold $\delta_{\text{soc}}(\mathcal{B}_{\varepsilon})$ is precisely the number of samples that an $\varepsilon$-quality initial point $X_0$ is worth, up to some conversion factor. But two major difficulties remain in this line of reasoning. First, evaluating $\delta_{\text{soc}}(X)$ for some $X \in \mathbb{R}^{n \times r}$ requires solving a minimization problem over the set of $\delta$-RIP operators. Second, evaluating $\delta_{\text{soc}}(\mathcal{B}_{\varepsilon})$ in turn requires minimizing $\delta_{\text{soc}}(X)$ over all choices of $X$ within an $\varepsilon$-neighborhood. Regarding the first point, Zhang et al. [23] showed that $\delta_{\text{soc}}(X)$ is the optimal value to a *convex* optimization problem, and can therefore be evaluated to arbitrary precising using a numerical algorithm. In the rank-1 case, they solved this convex optimization in closed-form, and use it to optimize over all $X \in \mathcal{B}_{\varepsilon}$. Their closed-form solution spanned 9 journal pages, and evoked a number of properties specific to the rank-1 case (for example, $xy^T + yx^T = 0$ implies $x = 0$ and $y = 0$, but $XY^T + YX^T = 0$ may hold for $X \neq 0$ and $Y \neq 0$). The authors noted that a similar closed-form solution for the general rank-$r$ case appeared exceedingly difficult. While overall proof technique is sharp and descriptive, its applicability appears to be entirely limited to the rank-1 case.

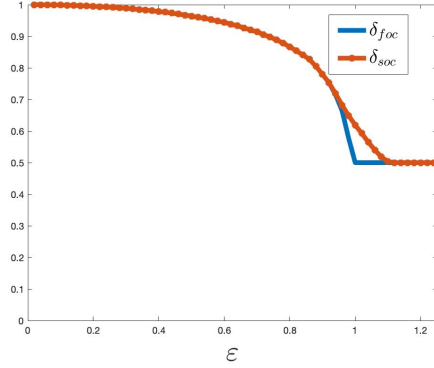

Figure 1: This paper is motivated by two key insights. First, it is relatively straightforward to solve $\delta_{\text{foc}}(X)$ in closed-form (Theorem 4). Second, the resulting lower-bound $\delta_{\text{soc}}(X) \geq \max\{\delta_{\text{foc}}(X), \delta^*\}$ ($\delta^* = 1/2$ for rank 1 and $\delta^* = 1/5$ for rank $> 1$) is remarkably tight. This means that $\max\{\delta_{\text{foc}}(\mathcal{B}_\varepsilon), \delta^*\}$ is a tight lower bound for $\delta_{\text{foc}}(\mathcal{B}_\varepsilon)$.

## 4  Main results

In this paper, we bypass the difficulty of deriving a closed-form solution for $\delta_{\text{soc}}(X)$ altogether by adopting a *sharp* lower-bound. This is based on two key insights. First, a spurious local minimum must also be a spurious critical point, so the analogous threshold over critical points would give an obvious lower-bound $\delta_{\text{foc}}(X) \leq \delta_{\text{soc}}(X)$.

**Definition 3** (Threshold for first-order condition)**.** Fix $Z \in \mathbb{R}^{n \times r}$. For $X \in \mathbb{R}^{n \times r}$, if $XX^T = ZZ^T$, then define $\delta_{\text{foc}}(X) = 1$. Otherwise, if $XX^T \neq ZZ^T$, then define

$$\delta_{\text{foc}}(X) \equiv \min_{\mathcal{A}}\{\delta : \nabla f_{\mathcal{A}}(X) = 0, \quad \mathcal{A} \text{ satisfies } \delta\text{-RIP}\}, \tag{5}$$

where the minimum is taken over all linear measurements $\mathcal{A} : \mathbb{R}^{n \times n} \to \mathbb{R}^m$. For $\mathcal{W} \subseteq \mathbb{R}^{n \times r}$, define $\delta_{\text{foc}}(\mathcal{W}) = \inf_{X \in \mathcal{W}} \delta_{\text{foc}}(X)$.

Whereas the main obstacle in Zhang et al. [23] is the considerable difficulty in deriving a closed-form solution for $\delta_{\text{soc}}(X)$, we show in this paper that it is relatively straightforward to solve $\delta_{\text{foc}}(X)$ in closed-form, to result in a simple, geometric solution.

**Theorem 4.** *Fix $Z \in \mathbb{R}^{n \times r}$. Given $\mathcal{A}$ satisfying $\delta$-RIP and $X \in \mathbb{R}^{n \times r}$ such that $XX^T \neq ZZ^T$, we have $\delta_{\text{foc}}(X) = \cos\theta$, where*

$$\sin\theta = \|Z^T(I - XX^\dagger)Z\|_F \, / \, \|XX^T - ZZ^T\|_F. \tag{6}$$

*and $X^\dagger$ denotes the pseudo-inverse of $X$. It follows that if $\delta < \cos\theta$, then $X$ is not a spurious critical point of $f_{\mathcal{A}}(X)$. If $\delta \geq \cos\theta$, then there exists some $\mathcal{A}^\star$ satisfying $\cos\theta$-RIP such that $\nabla f_{\mathcal{A}}(X) = 0$.*

The complete proof of Theorem 4 is given in Appendix A and a sketch is given in section 5. There is a nice geometric interpretation: the exact value of $\delta_{\text{foc}}(X)$ depends largely on the *incidence angle* between the column space of $X$ and the column space of $Z$. When the angle between $XX^T$ and $ZZ^T$ becomes small, the projection of $XX^T$ onto $ZZ^T$ becomes large. As a result, $\sin\theta$ becomes small and $\cos\theta$ becomes large. Therefore, Theorem 4 says that in regions where $XX^T$ and $ZZ^T$ are more aligned, fewer samples are required to prevent $X$ from becoming a spurious critical point. In regions where $XX^T$ and $ZZ^T$ are more orthogonal, a larger sample complexity is needed. Indeed, these are precisely the adversarial locations for which a large number of samples are required to prevent spurious local minima from appearing.

The lower-bound $\delta_{\text{foc}}(X) \leq \delta_{\text{soc}}(X)$ appears conservative, because critical points should be much more ubiquitous than local minima over a non-convex landscape. In particular, observe that $\delta_{\text{foc}}(X) = \cos\theta \to 0$ as $X \to 0$, which makes sense because $X = 0$ is a saddle point for all choices of $\mathcal{A}$. In other words, for any region $\mathcal{W}$ that contains 0, the lower-bound becomes trivial, as in $\delta_{\text{foc}}(\mathcal{W}) = 0 < \delta_{\text{soc}}(\mathcal{W})$. Our second insight here is that we must simultaneously have $\delta_{\text{soc}}(X) \geq 1/5$ due to

the global threshold of Bhojanapalli et al. [2] (or $\delta_{\mathrm{soc}}(x) \geq 1/2$ in the rank-1 case due to Zhang et al. [23]). Extending this idea over sets yields the following lower-bound

$$\delta_{\mathrm{soc}}(\mathcal{W}) \geq \max\{\delta_{\mathrm{foc}}(\mathcal{W}), \delta^*\} \qquad \text{for all } \mathcal{W} \subseteq \mathbb{R}^{n \times r}, \tag{7}$$

where $\delta^* = 1/2$ for $r = 1$ and $\delta^* = 1/5 > 1$. This bound is *remarkably* tight, as shown in Figure 1 for $\mathcal{W} = \mathcal{B}_\varepsilon$ over a range of $\varepsilon$. Explicitly solving the optimization $\delta_{\mathrm{foc}}(\mathcal{B}_\varepsilon) = \inf_{X \in \mathcal{B}_\varepsilon} \delta_{\mathrm{foc}}(X)$ using Theorem 4 and substituting into (7) yields the following.[1]

**Theorem 5.** *Let $\mathcal{A}$ satisfy $\delta$-RIP. Then we have $\delta_{\mathrm{foc}}(\mathcal{B}_\varepsilon) > \sqrt{1 - C\varepsilon}$ for all $\epsilon \leq 1/C$, where $C = \|ZZ^T\|_F / \sigma_{\min}^2(Z)$. Hence, if*

$$\delta < \max\left\{ \sqrt{[1 - C\varepsilon]_+}, \delta^* \right\} \tag{8}$$

*where $\delta^* = 1/2$ if $r = 1$ and $\delta^* = 1/5$ if $r > 1$, then $f_\mathcal{A}(X)$ has no spurious critical point within an $\varepsilon$-neighborhood of the solution:*

$$\nabla f_\mathcal{A}(X) = 0, \quad \|XX^T - ZZ^T\|_F \leq \varepsilon \|ZZ^T\|_F \quad \Longleftrightarrow \quad XX^T = ZZ^T. \tag{9}$$

The complete proof of this theorem is in Appendix B. Theorem 5 says that the number of samples needed to eliminate spurious critical points within an $\varepsilon$-neighborhood of the solution decreases dramatically as $\varepsilon$ becomes small. Given that $m \geq C_0 nr/\delta^2$ sub-Gaussian measurements are needed to satisfy $\delta$-RIP, we can translate Theorem 5 into the following sample complexity bound.

**Corollary 6.** *Let $\mathcal{A} : \mathbb{R}^{n \times n} \to \mathbb{R}^m$ be a sub-Gaussian measurement ensemble. If*

$$m \geq \min\left\{ \frac{1}{[1 - C\varepsilon]_+}, 25 \right\} C_0 nr$$

*then with high probability there are no spurious local minima within $\mathcal{B}_\varepsilon$.*

The proof of Corollary 6 follows immediately from Theorem 5 combined with the direct relationship between the RIP-property and the sample complexity for sub-Gaussian measurement ensembles. We see that the relationship between the quality of the initial point and the number of samples saved is essentially *linear*. Improving the quality of the initial point by a linear factor corresponds to a linear decrease in sample complexity. Moreover, the rate of improvement depends on the constant $C$. This shows that in the non-convex setup of matrix sensing, there is a significant difference between a good initial point and a mediocre initial point. In the case that $C = \|ZZ^T\|_F / \sigma_{\min}^2(Z)$ is large, this difference is even more pronounced.

## 5 Proof of Main Results

### 5.1 Notation and Definitions

We use $\| \cdot \|$ for the vector 2-norm and use $\| \cdot \|_F$ to denote the Frobenius norm of a matrix. For two square matrices $A$ and $B$, $A \succeq B$ means $B - A$ is positive semidefinite. The trace of a square matrix $A$ is denoted by $\mathrm{tr}(A)$. The *vectorization* $\mathrm{vec}(A)$ is the length-$mn$ vector obtained by stacking the columns of $A$. Let $\mathcal{A} : \mathbb{R}^{n \times n} \to \mathbb{R}^m$ be a linear measurement operator, and let $Z \in \mathbb{R}^{n \times r}$ be a fixed ground truth matrix. We define $\mathbf{A} = [\mathrm{vec}(A_1), \ldots, \mathrm{vec}(A_m)]$ as the matrix representation of $\mathcal{A}$, and note that $\mathrm{vec}[\mathcal{A}(X)] = \mathbf{A}\,\mathrm{vec}(X)$. We define the error vector $\mathbf{e}$ and its Jacobian $\mathbf{X}$ to satisfy

$$\mathbf{e} = \mathrm{vec}(XX^T - ZZ^T) \tag{10a}$$

$$\mathbf{X}\,\mathrm{vec}(Y) = \mathrm{vec}(XY^T + YX^T) \qquad \text{for all } Y \in \mathbb{R}^{n \times r}. \tag{10b}$$

### 5.2 Proof Sketch of Theorem 4

A complete proof of Theorem 4 relies on a few technical lemmas, so we defer the complete proof to Appendix A. The key insight is that $\delta_{\mathrm{foc}}(X)$ is the solution to a *convex* optimization problem, which we can solve in closed-form. At first sight, evaluating $\delta_{\mathrm{foc}}(X)$ seems very difficult as it involves solving an optimization problem over the set of $\delta$-RIP operators, as defined in equation 5 . However,

a minor modification of Theorem 8 in Zhang et al. [23] shows that $\delta_{\text{foc}}(X)$ can be reformulated as a convex optimization problem of the form

$$\eta(X) \quad \equiv \quad \max_{\eta, \mathbf{H}} \left\{ \eta \quad : \quad \mathbf{X}^T \mathbf{H} \mathbf{e} = 0, \quad \eta I \preceq \mathbf{H} \preceq I \right\}. \tag{11}$$

where $\eta(X)$ is related to $\delta_{\text{foc}}(X)$ by

$$\delta_{\text{foc}}(X) = \frac{1 - \eta(X)}{1 + \eta(X)}. \tag{12}$$

We will show that problem (11) actually has a simple closed-form solution. First, we write its Lagrangian dual as

$$
\begin{aligned}
\underset{y, U_1, U_2}{\text{minimize}} \quad & \text{tr}(U_2) \\
\text{subject to} \quad & (\mathbf{X}y)\mathbf{e}^T + \mathbf{e}(\mathbf{X}y)^T = U_1 - U_2 \\
& \text{tr}(U_1) = 1, \quad U_1, U_2 \succeq 0.
\end{aligned}
\tag{13}
$$

Notice that strong duality holds because Slater's condition is trivially satisfied by the dual: $y = 0$ and $U_1 = U_2 = 2I/n(n+1)$ is a strictly feasible point. It turns out that the dual problem can be rewritten as an optimization problem over the eigenvalues of the matrix $(\mathbf{X}y)\mathbf{e}^T + \mathbf{e}(\mathbf{X}y)^T$. The proof of this in in Appendix A.

For any $\alpha \in \mathbb{R}$ we denote $[\alpha]_+ = \max\{0, +\alpha\}$ and $[\alpha]_- = \max\{0, -\alpha\}$. The dual problem can be written as

$$\min_y \frac{\text{tr}[M(y)]_-}{\text{tr}[M(y)]_+} = \min_y \frac{\sum_i \lambda_i [M(y)]_-}{\sum_i \lambda_i [M(y)]_+}, \quad \text{where} \quad M(y) = (\mathbf{X}y)\mathbf{e}^T + \mathbf{e}(\mathbf{X}y)^T,$$

and $\lambda_i[M(y)]$ denotes the eigenvalues of the rank-2 matrix $M(y)$. It is easy to verify that the only two non-zero eigenvalues of $(\mathbf{X}y)\mathbf{e}^T + \mathbf{e}(\mathbf{X}y)^T$ are

$$\|\mathbf{X}y\| \|\mathbf{e}\| \left(\cos\theta_y \pm 1\right), \quad \text{where } \cos\theta_y = \frac{\mathbf{e}^T \mathbf{X}y}{\|\mathbf{e}\| \|\mathbf{X}y\|}.$$

It follows that

$$\eta(X) = \min_y \frac{1 - \cos\theta_y}{1 + \cos\theta_y}$$

and therefore

$$\delta_{\text{foc}}(X) = \max_y \cos\theta_y = \max_y \frac{\mathbf{e}^T \mathbf{X}y}{\|\mathbf{e}\| \|\mathbf{X}y\|}.$$

Let $y^*$ be the optimizer of the optimization problem above, then $\theta_{y^*}$ is simply the incidence angle between the column space of $X$ and the error vector $\mathbf{e}$. Thus we have $y^\star = \arg\min_y \|\mathbf{e} - \mathbf{X}y\|$. Using Lemma 10 in Appendix A, we show that solving for $y^*$ yields a closed-form expression for $\theta_{y^*}$ in the form

$$\sin\theta_{y^*} = \frac{\|Z^T(I - XX^\dagger)Z\|_F}{\|XX^T - ZZ^T\|_F}.$$

Hence we have $\delta_{\text{foc}}(X) = \cos\theta$, with $\theta = \theta_{y^*}$ given by the equation above.

### 5.3   Proof of Theorem 5

The proof of Theorem 5 is based on the following lemma. Its proof is very technical and can be can be found in Appendix B.

**Lemma 7.** *Let $Z \neq 0$ and suppose that $\|XX^T - ZZ^T\|_F \leq \epsilon \|ZZ\|_F^2$. Then*

$$\sin^2\theta = \frac{\|Z^T(I - XX^\dagger)Z\|_F^2}{\|XX^T - ZZ^T\|_F^2} \leq \frac{\epsilon}{2\sigma_{\min}^2(Z)/\|ZZ^T\|_F - \epsilon}.$$

To prove Theorem 5, we simply set $C_1 = \sigma^2_{\min}(Z)/\|ZZ^T\|_F$ and write

$$\cos\theta = \sqrt{1 - \sin^2\theta} \geq \sqrt{1 - \frac{\epsilon}{2C_1 - \epsilon}}.$$

It is easy to see that $\frac{\epsilon}{2C_1-\epsilon}$ is dominated by the linear function $\varepsilon/C_1$ so long as $\epsilon \leq C_1$. This follows directly from the fact that $\frac{\epsilon}{2C_1-\epsilon}$ is convex between $0$ and $C_1$. Thus we have

$$\cos\theta \geq \sqrt{1 - \frac{\epsilon}{C_1}}$$

Since this lower bound holds for all $X$ in $\mathcal{B}_\varepsilon$, it follows that $\delta_{\text{foc}}(\mathcal{B}_\varepsilon) \geq \sqrt{1 - \varepsilon/C_1}$.

## 6  Numerical Results

In this section we give a geometric interpretation for Theorem 4, which we already alluded to in section 4: the sample complexity to eliminate spurious critical points is small in regions where the column spaces of $X$ and $Z$ are more aligned and large in regions where they are orthogonal. We also numerically verify that $\delta_{\text{foc}}(X)$ is a tight lower bound for $\delta_{\text{soc}}(X)$ for a wide range of $\varepsilon$, providing numerical evidence that the bound in Theorem 5 is tight.

Our main results and geometric insights hold for *any rank*, but for ease of visualization we focus on the rank-1 case where $x$ and $z$ are now just vectors. To measure the alignment between the column space of $x$ and that of $z$ in the rank-1 case, we define the length ratio and the incidence angle as

$$\rho = \frac{\|x\|}{\|z\|}, \quad \cos\phi = \frac{x^T z}{\|x\|\|z\|}.$$

Our goal is to plot how sample complexity depends on this alignment. Visualizing the dependence of sample complexity on $\rho$ and $\cos\phi$ is particularly easy in rank-1 because these two parameters completely determine the values of both $\delta_{\text{foc}}(x)$ and $\delta_{\text{soc}}(x)$. See [23] section 8.1 for a proof of this fact. This allows us to plot the level curves of $\delta_{\text{foc}}(x)$ and $\delta_{\text{soc}}(x)$ over the parameter space $\rho$ and $\phi$ in Figure 2. This is shown by the blue curves. Since we are particularly interested in sample complexity near the ground truth, we also plot the level sets of the function $\|xx^T - zz^T\|_F/\|zz^T\|_F$ using red curves. The horizontal axis is the value of $\rho\cos\phi$ and the vertical axis is the value of $\rho\sin\phi$.

We can immediately see that in regions in the optimization landscape where $x$ is more aligned with $z$, i.e., when $\sin\phi$ is small, the values of both threshold functions tend to be high and a relatively small number of samples suffices to prevent $x$ from becoming a spurious critical point. However, when $x$ and $z$ becomes closer to being orthogonal, i.e., when $\cos\phi$ is close to 0, then $\delta_{\text{foc}}(x)$ becomes arbitrarily small, and $\delta_{\text{soc}}(x)$ also becomes smaller, albeit to a lesser extent. As a result, preventing $x$ from becoming a spurious critical point (or spurious local minima) in these regions require many more samples. This intuition also permeates to the high-rank case, even though visualization becomes difficult, and a slightly more general definition of length ratio and alignment is required. Similar to the rank-1 case, in regions where $XX^T$ and $ZZ^T$ are more aligned, the sample complexity required to eliminate spurious critical points is small and in regions where $XX^T$ and $ZZ^T$ are close to orthogonal, a small sample complexity is required.

Regarding the tightness of using $\delta_{\text{foc}}(X)$ as a lower bound for $\delta_{\text{soc}}(X)$, note that if we look at the level sets of $\|xx^T - zz^T\|_F/\|zz^T\|_F$, we see that in regions close to the ground truth, both $\delta_{\text{soc}}(x)$ and $\delta_{\text{foc}}(x)$ are very close to 1. This is in perfect agreement with our results in Theorem 5, where we showed that a small $\varepsilon$ results in a large $\delta_{\text{foc}}(\mathcal{B}_\varepsilon)$. Moreover, the shapes of the level curves of $\delta_{\text{soc}}$ and $\delta_{\text{foc}}$ that flow through the regions near the ground truth are almost identical. This indicates that for a large region near the ground truth, the second-order condition, i.e., the hessian being positive semidefinite, is inactive. This is the underlying mechanism that causes $\delta_{\text{foc}}$ to be a tight lower bound for $\delta_{\text{foc}}$.

## 7  Conclusions

Recent work by Bhojanapalli et al. [2] has shown that the non-convex optimization landscape of matrix sensing contains no spurious local minima when there are sufficiently large amount of samples.

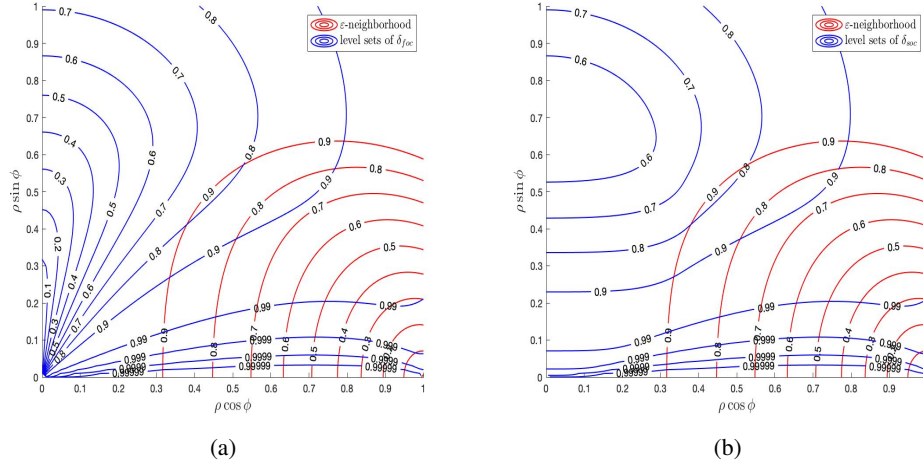

Figure 2: (a) the level sets of $\delta_{\mathrm{foc}}$ and $\|xx^T - zz^T\|_F/\|zz^T\|_F$ (b) the level sets of $\delta_{\mathrm{soc}}$ and $\|xx^T - zz^T\|_F/\|zz^T\|_F$

However, these theoretical bounds on the sample complexity are very conservative compared to the number of samples needed in real applications like power state estimation. In our paper, we provide one explanation for this phenomenon: in real life, we often have access to good initial points, which can reduce the number of samples we need. The main results of our paper give a mathematical characterization of this phenomenon. We define a function $\delta_{\mathrm{soc}}(X)$ that gives a *precise* threshold on the number of samples needed to prevent $X$ from becoming a spurious local minima. Although $\delta_{\mathrm{soc}}$ is difficult to compute exactly, we obtain a closed-form, sharp lower bound using convex optimization. As a result, we are able to characterize the *tradeoff* between the quality of the initial point and the sample complexity. In particular, we show that a linear improvement in the quality of the initial point corresponds to a linear decrease in sample complexity.

On a more general level, our work uses new techniques to paint a full picture for the non-convex landscape of matrix sensing: the problem becomes more "non-convex" (requiring more samples to eliminate spurious local minima) as we get further and further away from the global min. Once we are sufficiently far away, it becomes necessary to rely on global guarantees instead. Thus, our work brings new insight into how a non-convex problem can gradually become more tractable either through more samples or a better initial point and provides a tradeoff between these two mechanisms. For future work, it would be interesting to see if similar techniques can be extended to other non-convex models such as neural networks.

## Acknowledgements

Partial financial support was provided by the National Science Foundation under award ECCS-1808859.

## Broader Impact

Many modern applications in engineering and computer science, and in machine learning in particular often have to deal with non-convex optimization. However, many aspects of non-convex optimization are still not well understood. Our paper provides more insight into the optimization landscape of a particular problem: low-rank matrix factorization. In addition, the methods we develop can potentially be used to understand many other non-convex problems. This is a step towards a more thorough analysis of current algorithms for non-convex optimization and also a step towards developing better and more efficient algorithms with theoretical guarantees.

## Footnotes

[1] We denote $[x]_+ = \max\{0, x\}$.

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
