[Supplementary Material]

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

# Appendix A

In Appendix A we fill out the missing details in the proof sketch of Section 5.2 and provide a complete proof of theorem 4, which we restate below as theorem 8. The main idea behind theorem 4 was that we can find a simple *closed* form solution for the first order threshold function, which was defined as the solution to the following optimization problem:

$$\delta_{\text{foc}}(X) \equiv \min_{\mathcal{A}}\{\delta : \nabla f_{\mathcal{A}}(X) = 0, \quad \mathcal{A} \text{ satisfies } \delta\text{-RIP}\} \tag{14}$$

**Theorem 8.** *(Same as theorem 4). Fix $Z \in \mathbb{R}^{n \times r}$. Given $\mathcal{A}$ satisfying $\delta$-RIP and $X \in \mathbb{R}^{n \times r}$ such that $XX^T \neq ZZ^T$, we have $\delta_{\text{foc}}(X) = \cos\theta$, where*

$$\sin\theta = \|Z^T(I - XX^\dagger)Z\|_F \,/\, \|XX^T - ZZ^T\|_F. \tag{15}$$

*and $X^\dagger$ denotes the pseudo-inverse of $X$. It follows that if $\delta < \cos\theta$, then $X$ is not a spurious critical point of $f_{\mathcal{A}}(X)$. If $\delta \geq \cos\theta$, then there exists some $\mathcal{A}^\star$ satisfying $\cos\theta$-RIP such that $\nabla f_{\mathcal{A}}(X) = 0$.*

In (14), the value of $\delta_{\text{foc}}(X)$ is defined as an optimization problem over measurement operators that satisfy $\delta$-RIP. While this problem in non-convex, Zhang et al.[23] showed that the convex-relaxation of this problem is actually exact. In particular, using theorem 8 from [23], the optimization problem (14) can be formulated as

$$\eta(X) \quad \equiv \quad \max_{\eta, \mathbf{H}}\left\{\eta \quad : \quad \mathbf{X}^T\mathbf{H}\mathbf{e} = 0, \quad \eta I \preceq \mathbf{H} \preceq I\right\}. \tag{16}$$

where $\eta(X) = (1 - \delta_{\text{foc}}(X))/(1 + \delta_{\text{foc}}(X))$. Notice that (16) is just a semidefinite program in the variables $\eta$ and $\mathbf{H}$. Our goal is to find a closed-form solution for this SDP, which will immediately yield a closed-form solution for $\delta_{\text{foc}}(X)$.

Before we prove the theorem 8, we first prove two technical lemmas. The first lemma gives an explicit solution to the eigenvalues of a rank-2 matrix and the second lemma characterizes the solution to an SDP that will be a part of the proof of theorem 4.

**Lemma 9.** *Given $a, b \in \mathbb{R}^n$, the matrix $M = ab^T + ba^T$ has eigenvalues $\lambda_1 \geq \cdots \geq \lambda_n$ where:*

$$\lambda_i = \begin{cases} +\|a\|\|b\|(1 + \cos\theta) & i = 1 \\ -\|a\|\|b\|(1 - \cos\theta) & i = n \\ 0 & \text{otherwise} \end{cases}$$

*and $\theta \equiv \arccos\left(\frac{a^T b}{\|a\|\|b\|}\right)$ is the angle between $a$ and $b$.*

*Proof.* Without loss of generality, assume that $\|a\| = \|b\| = 1$. (Otherwise, we can rescale $\hat{a} = a/\|a\|, \hat{b} = b/\|b\|$ and write $M = \|a\|\|b\|(\hat{a}\hat{b}^T + \hat{b}\hat{a}^T)$. Now decompose $b$ into a tangent and normal component with respect to $a$, as in

$$b = a\underbrace{a^T b}_{\cos\theta} + \underbrace{(I - aa^T)b}_{c\sin\theta} = a\cos\theta + c\sin\theta$$

where $c$ is a unit normal vector with $\|c\| = 1$ and $a^T c = 0$. Thus $ab^T + ba^T$ can be written as

$$ab^T + ba^T = \begin{bmatrix} a & c \end{bmatrix}\begin{bmatrix} 2\cos\theta & \sin\theta \\ \sin\theta & 0 \end{bmatrix}\begin{bmatrix} a & c \end{bmatrix}^T.$$

This shows that $M$ is spectrally similar to a $2 \times 2$ matrix with eigenvalues $\cos\theta \pm 1$. $\qquad\square$

**Lemma 10.** *Given a matrix $M \neq 0$ we can split the matrix $M$ into a positive and negative part satisfying*

$$M = M_+ - M_- \quad \text{where} \quad M_+, M_- \succeq 0, \quad M_+M_- = 0.$$

*Then the following problem has solution*

$$\min_{\substack{\alpha \in \mathbb{R} \\ U, V \succeq 0}} \{\text{tr}(V) : \text{tr}(U) = 1, \alpha M = U - V\} = \min\left\{\frac{\text{tr}(M_-)}{\text{tr}(M_+)}, \frac{\text{tr}(M_+)}{\text{tr}(M_-)}\right\}.$$

*Proof.* In this proof we will consider two cases: $\mathrm{tr}(M_-) \leq \mathrm{tr}(M_+)$ and $\mathrm{tr}(M_-) \geq \mathrm{tr}(M_+)$. We'll see that in the first case, the optimal value is $\mathrm{tr}(M_-)/\mathrm{tr}(M_+)$ and in the second case, the optimal value is $\mathrm{tr}(M_+)/\mathrm{tr}(M_1)$.

First, assume that $\mathrm{tr}(M_-) \leq \mathrm{tr}(M_+)$. Let $p^*$ be the optimal value. Then we have

$$p^\star = \max_{\beta} \min_{\substack{\alpha \in \mathbb{R} \\ U, V \succeq 0}} \{\mathrm{tr}(V) + \beta \cdot [1 - \mathrm{tr}(U)] : \alpha M = U - V\} \tag{17}$$

$$= \max_{\beta} \min_{\alpha \in \mathbb{R}} \left\{ \beta + \min_{U,V \succeq 0} \{\mathrm{tr}(V) - \beta \cdot \mathrm{tr}(U) : \alpha M = U - V\} \right\}$$

$$= \max_{\beta} \min_{\alpha \in \mathbb{R}} \left\{ \beta + \min_{U} \left[ \mathrm{tr}\left( U - \alpha M \right) - \beta \cdot \mathrm{tr}\left( U \right) \right] : U - \alpha M \succeq 0, U \succeq 0 \right\}$$

$$= \max_{\beta} \min_{\alpha \in \mathbb{R}} \left\{ \beta + \min_{U} [-\alpha \mathrm{tr}(M) + (1 - \beta)\mathrm{tr}(U)] : U - \alpha M \succeq 0, U \succeq 0 \right\}$$

$$= \max_{\beta \leq 1} \min_{\alpha \in \mathbb{R}} \left\{ \beta + \min_{U} [-\alpha \mathrm{tr}(M) + (1 - \beta)\mathrm{tr}(U)] : U - \alpha M \succeq 0, U \succeq 0 \right\}. \tag{18}$$

Note that the first line converts the equality constraint into a Lagrangian. The second line simply rearranges the terms. The third line plugs in $V = U - \alpha M$. The fourth line again rearranges the terms. The last line follows from the observation that if $\beta > 1$, then the inner minimization over $U$ will go to negative infinity since the trace of $U$ can be arbitrarily large.

First, consider the case $\alpha \geq 0$. Then we have $\alpha M = \alpha M_+ - \alpha M_-$. Since $1 - \beta \geq 0$, the minimization over $U$ is achieved at $U = \alpha M_+$. Plugging this value into the optimization problem, then (19) becomes

$$\max_{\beta \leq 1} \min_{\alpha \geq 0} \{\beta + \alpha[\mathrm{tr}(M_-) - \beta \mathrm{tr}(M_+)]\}$$

If $\mathrm{tr}(M_-) - \beta \mathrm{tr}(M_+) < 0$, then the optimal value of the inner minimization will go to negative infinity. On the other hand, if $\mathrm{tr}(M_-) - \beta \mathrm{tr}(M_+) \geq 0$ then the minimum inside is achieved at $\alpha = 0$. Thus the problem above is equivalent to

$$\max_{\beta \leq 1} \{\beta : \mathrm{tr}(M_-) - \beta \mathrm{tr}(M_+) \geq 0\}.$$

Since $\mathrm{tr}(M_-) \leq \mathrm{tr}(M_+)$, the optimal value of the problem above is achieved at $\mathrm{tr}(M_-)/\mathrm{tr}(M_+) \leq 1$. Now suppose that $\alpha \leq 0$. Then the optimal value for $U$ is achieved at $U = -\alpha M_-$. Plugging this value in and (19) becomes

$$\max_{\beta \leq 1} \min_{\alpha \leq 0} \{\beta + \alpha[\beta \mathrm{tr}(M_-) - \mathrm{tr}(M_+)]\}.$$

Similar to before, we must have $\beta \mathrm{tr}(M_-) - \mathrm{tr}(M_+) \leq 0$, so $\beta \leq \mathrm{tr}(M_+)/\mathrm{tr}(M_-)$. Since $\mathrm{tr}(M_-) \leq \mathrm{tr}(M_+)$, the optimal value in this case is just $\beta = 1$. Combining the results for $\alpha \geq 0$ and $\alpha \leq 0$, we find that when $\mathrm{tr}(M_-) \leq \mathrm{tr}(M_+)$, the optimal value is

$$p^* = \min \left\{ 1, \frac{\mathrm{tr}(M_-)}{\mathrm{tr}(M_+)} \right\} = \frac{\mathrm{tr}(M_-)}{\mathrm{tr}(M_+)}.$$

Repeating the same arguments for when $\mathrm{tr}(M_-) \geq \mathrm{tr}(M_+)$, we see that in this case the optimal value becomes

$$p^* = \min \left\{ \frac{\mathrm{tr}(M_+)}{\mathrm{tr}(M_-)}, 1 \right\} = \frac{\mathrm{tr}(M_+)}{\mathrm{tr}(M_-)}.$$

Finally, combining these two cases, i.e., $\mathrm{tr}(M_-) \geq \mathrm{tr}(M_+)$ and $\mathrm{tr}(M_-) \leq \mathrm{tr}(M_+)$, we obtain

$$p^* = \min \left\{ \frac{\mathrm{tr}\left( M_- \right)}{\mathrm{tr}\left( M_+ \right)}, \frac{\mathrm{tr}\left( M_+ \right)}{\mathrm{tr}\left( M_- \right)} \right\},$$

which completes the proof. $\qquad\square$

Using the previous two lemmas, we can now write out a closed-form solution for (16). The main idea is to solve the dual problem of (16) instead, and since strong duality holds, this will also yield a solution to the primal problem. Recall that in Section 5.2, we wrote the dual of problem (16) as

$$\min_{y, U_1, U_2} \quad \mathrm{tr}(U_2) \tag{19}$$

$$\text{subject to} \quad (\mathbf{X}y)\mathbf{e}^T + \mathbf{e}(\mathbf{X}y)^T = U_1 - U_2$$
$$\mathrm{tr}(U_1) = 1, \quad U_1, U_2 \succeq 0.$$

Notice that here strong duality holds because Slater's condition is trivially satisfied by the dual: $y = 0$ and $U_1 = U_2 = 2I/n(n+1)$ is a strictly feasible point.

We claim that this dual problem can be rewritten as an optimization problem over the eigenvalues of a rank-2 matrix. This is given in the lemma below. To simplify notation, here we define a positive/negative splitting: for any $\alpha \in \mathbb{R}_+$ we denote $[\alpha]_+ = \max\{0, +\alpha\}$ and $[\alpha]_- = \max\{0, -\alpha\}$. This idea can be extended to matrices by applying splitting to the eigenvalues.

**Lemma 11.** *Given data* $\mathbf{e}$ *and* $\mathbf{X} \neq 0$*, define*

$$\eta = \min_{y, U_1, U_2} \quad \mathrm{tr}(U_2) \tag{20}$$
$$\text{subject to} \quad (\mathbf{X}y)\mathbf{e}^T + \mathbf{e}(\mathbf{X}y)^T = U_1 - U_2$$
$$\mathrm{tr}(U_1) = 1, \quad U_1, U_2 \succeq 0.$$

*Define* $M(y)$ *to be the rank-2 matrix* $(\mathbf{X}y)\mathbf{e}^T + \mathbf{e}(\mathbf{X}y)^T$ *and let* $\lambda_i[M(y)]$ *denote its eigenvalues. Then* $\eta$ *can be evaluated as*

$$\eta = \min_{y \neq 0} \frac{\mathrm{tr}[M(y)]_-}{\mathrm{tr}[M(y)]_+} = \min_{y \neq 0} \frac{\sum_i \lambda_i[M(y)]_-}{\sum_i \lambda_i[M(y)]_+} = \min_{y \neq 0} \frac{1 - \cos\theta_y}{1 + \cos\theta_y},$$

*where* $\cos\theta_y = \mathbf{e}^T \mathbf{X}y / \|\mathbf{e}\|\|\mathbf{X}y\|$.

The proof of Lemma 11 relies mainly on the two lemmas we proved in the preceding section.

*Proof.* Let $y = \alpha\hat{y}$, where $\|\hat{y}\| = 1$ and $\alpha \in \mathbb{R}^n$. Thus the optimization problem (20) becomes

$$\eta = \min_{\alpha, \hat{y}, U_1, U_2} \quad \mathrm{tr}(U_2)$$
$$\text{subject to} \quad \alpha \cdot [(\mathbf{X}\hat{y})\mathbf{e}^T + \mathbf{e}(\mathbf{X}\hat{y})^T] = U_1 - U_2$$
$$\mathrm{tr}(U_1) = 1, \quad \|\hat{y}\| = 1, \quad U_1, U_2 \succeq 0.$$

To solve this problem, first we keep $\hat{y}$ fixed, and optimize over $\alpha, U_1, U_2$. This gives us the problem

$$\min_{\alpha, U_1, U_2} \quad \mathrm{tr}(U_2)$$
$$\text{subject to} \quad \alpha \cdot [(\mathbf{X}\hat{y})\mathbf{e}^T + \mathbf{e}(\mathbf{X}\hat{y})^T] = U_1 - U_2$$
$$\mathrm{tr}(U_1) = 1, \quad U_1, U_2 \succeq 0.$$

Notice that if we set $M(\hat{y}) = (\mathbf{X}\hat{y})\mathbf{e}^T + \mathbf{e}(\mathbf{X}\hat{y})^T$, then the problem above is in exactly the same form as the one in lemma 10. Therefore, its optimal value is

$$\min \left\{ \frac{\mathrm{tr}\,(M(\hat{y})_-)}{\mathrm{tr}\,(M(\hat{y})_+)}, \frac{\mathrm{tr}\,(M(\hat{y})_+)}{\mathrm{tr}\,(M(\hat{y})_-)} \right\}.$$

Finally, to obtain $\eta$, we still need to optimize over $\hat{y}$, i.e.,

$$\eta = \min_{\|\hat{y}\|=1} \min \left\{ \frac{\mathrm{tr}\,(M(\hat{y})_-)}{\mathrm{tr}\,(M(\hat{y})_+)}, \frac{\mathrm{tr}\,(M(\hat{y})_+)}{\mathrm{tr}\,(M(\hat{y})_-)} \right\}.$$

Since both the numerator and the denominator are linear in $y$, we can ignore the constraint $\|\hat{y}\| = 1$ and simply optimize over $y$, which gives us

$$\eta = \min_{y \neq 0} \min \left\{ \frac{\mathrm{tr}\,(M(y)_-)}{\mathrm{tr}\,(M(y)_+)}, \frac{\mathrm{tr}\,(M(y)_+)}{\mathrm{tr}\,(M(y)_-)} \right\}.$$

With lemma 9, we see that the only two eigenvalues of $M(y)$ are

$$\|\mathbf{X}y\|\|y\|(\cos\theta_y + 1), \qquad \|\mathbf{X}y\|\|y\|(\cos\theta_y - 1),$$

where $\cos\theta_y = \mathbf{e}^T \mathbf{X}y / \|\mathbf{e}\|\|\mathbf{X}y\|$. It follows that $\mathrm{tr}(M_-) = \|\mathbf{X}y\|\|y\|(1 - \cos\theta_y)$ and $\mathrm{tr}(M_+) = \|\mathbf{X}y\|\|y\|(\cos\theta_y + 1)$. Thus

$$\eta = \min_{y \neq 0} \min \left\{ \frac{1 - \cos\theta_y}{1 + \cos\theta_y}, \frac{1 + \cos\theta_y}{1 - \cos\theta_y} \right\}.$$

Notice that in the optimization problem above, if the minimum is achieved at some $y^*$, it must also be achieved at $-y^*$, due to symmetry. Therefore, it suffices to optimize over only the first term $\frac{1-\cos\theta_y}{1+\cos\theta_y}$, so we get

$$\eta = \min_{y \neq 0} \frac{1 - \cos\theta_y}{1 + \cos\theta_y}.$$

This completes the proof. □

Notice that Lemma 11 reduces problem (16) to only depend on the values of $\cos\theta_y$. Now, to complete the proof of theorem 4, we just need one additional lemma that gives a closed form solution for $\cos\theta_y$, which we state below.

**Lemma 12.** *Let $X, Z$ be $n \times r$ matrices of any rank, and define $\mathbf{e}$ and $\mathbf{X} \neq 0$ as in equations 10(a) and 10(b). Then, the incidence angle $\theta$ between $\mathbf{e}$ and $\mathrm{range}(\mathbf{X})$, defined as in*

$$\cos\theta = \max_{y \neq 0} \left\{ \frac{\mathbf{e}^T \mathbf{X} y}{\|\mathbf{e}\| \|\mathbf{X} y\|} \right\} = \frac{\|\mathbf{X}\mathbf{X}^\dagger \mathbf{e}\|}{\|\mathbf{e}\|},$$

*has closed-form expression*

$$\sin\theta = \frac{\|Z^T(I - XX^\dagger)Z\|_F}{\|XX^T - ZZ^T\|_F}$$

*where $X^\dagger$ denotes the Moore–Penrose pseudoinverse of $X$.*

*Proof.* Define $y^\star = \arg\min_y \|\mathbf{e} - \mathbf{X}y\|$ and decompose $\mathbf{e} = \mathbf{X}y^\star + w$. The optimality condition for $y^\star$ reads $\mathbf{X}^T(\mathbf{e} - \mathbf{X}y^\star) = \mathbf{X}^T w = 0$, so we substitute $\mathbf{e}^T\mathbf{X} = (y^*)^T\mathbf{X}^T\mathbf{X}$ to yield

$$\|\mathbf{e}\|\cos\theta = \|\mathbf{e}\| \max_{y \neq 0} \left\{ \frac{\mathbf{e}^T\mathbf{X}y}{\|\mathbf{e}\|\|\mathbf{X}y\|} \right\} = \max_{y \neq 0} \left\{ \frac{(y^\star)^T\mathbf{X}^T\mathbf{X}y}{\|\mathbf{X}y\|} \right\} = \|\mathbf{X}y^\star\|,$$

and therefore $\|\mathbf{e}\|\sin\theta = \|w\| = \min_y \|\mathbf{e} - \mathbf{X}y\|$, because we have $\mathbf{e} = \mathbf{X}y^* + w$ with $w^T\mathbf{X}y^* = 0$. Now, define $Q = \mathrm{orth}(X) \in \mathbb{R}^{n \times q}$ where $q = \mathrm{rank}(X) \leq r$, and define $P \in \mathbb{R}^{n \times (n-q)}$ as the orthogonal complement of $Q$. Decompose $X = Q\hat{X}$, and $Z = Q\hat{Z}_1 + P\hat{Z}_2$, and note that

$$\|w\| = \min_y \|\mathbf{e} - \mathbf{X}y\|$$
$$= \min_Y \|(XX^T - ZZ^T) - (XY^T + YX^T)\|_F$$
$$= \min_{[\hat{Y}_1; \hat{Y}_2] \in \mathbb{R}^{n \times r}} \left\| \begin{bmatrix} \hat{X}\hat{X}^T - \hat{Z}_1\hat{Z}_1^T & -\hat{Z}_1\hat{Z}_2^T \\ -\hat{Z}_2\hat{Z}_1^T & -\hat{Z}_2\hat{Z}_2^T \end{bmatrix} - \begin{bmatrix} \hat{X}\hat{Y}_1^T + \hat{Y}_1\hat{X}^T & \hat{X}\hat{Y}_2^T \\ \hat{Y}_2\hat{X}^T & 0 \end{bmatrix} \right\|_F$$
$$= \|\hat{Z}_2\hat{Z}_2^T\|_F$$

From the second line to the third, we apply a change of basis onto $[Q\ P]$, which preserves the Frobenius norm. To derive the last line, notice that the $q \times r$ matrix $\hat{X}$ has full row rank, so that $\hat{X}\hat{X}^T \succ 0$ and $\hat{X}\hat{X}^\dagger = I_q$. We want to show that there exists $\hat{Y}_1$ such that

$$\hat{X}\hat{Y}_1^T + \hat{Y}_1\hat{X}^T = \hat{X}\hat{X}^T - \hat{Z}_1\hat{Z}_1^T.$$

Since the right hand side is symmetric, we can write it as $L + L^T$, where $L$ is some lower-triangular matrix. Thus it suffices to show that there exists $\hat{Y}_1$ such that $\hat{X}\hat{Y}_1^T = L$, which follows from that fact that $\hat{X}$ has full row-rank. Similarly, there exists some $\hat{Y}_2$ such that $\hat{X}\hat{Y}_2 = -\hat{Z}_2\hat{Z}_1^T$. Thus, all terms except the last one cancels out and we are left with $\min_y \|\mathbf{e} - \mathbf{X}y\| = \|\hat{Z}_2\hat{Z}_2^T\|_F$.

Finally, note that $Q\hat{Z}_1 = XX^\dagger Z$ and $P\hat{Z}_2 = (I - XX^\dagger)Z$ and that

$$\|\hat{Z}_2\hat{Z}_2^T\|_F^2 = \|P\hat{Z}_2\hat{Z}_2^T P^T\|_F^2$$
$$= \|(I - XX^\dagger)ZZ^T(I - XX^\dagger)\|_F^2$$
$$= \mathrm{tr}[(I - XX^\dagger)ZZ^T(I - XX^\dagger)ZZ^T(I - XX^\dagger)]$$
$$= \mathrm{tr}[Z^T(I - XX^\dagger)ZZ^T(I - XX^\dagger)Z]$$
$$= \|Z^T(I - XX^\dagger)Z\|_F^2.$$

Substituting the definition of $\mathbf{e}$ completes the proof. □

Now theorem 4 will be a direct consequence of lemma 11 and lemma 12. We give a proof below.

*Proof.* (Theorem 4). Note that $\delta_{\text{foc}}$ is related to $\eta$ by the equation

$$\eta = \frac{1 - \delta_{\text{foc}}}{1 + \delta_{\text{foc}}}.$$

Applying lemma 11, we immediately get

$$\delta_{\text{foc}}(X) = \max_{y \neq 0} \cos \theta_y = \max_{y \neq 0} \frac{\mathbf{e}^T \mathbf{X} y}{\|\mathbf{e}\| \|\mathbf{X} y\|}.$$

From lemma 12, we see that this optimization problem over $y$ has a simple closed form solution of the form

$$\delta_{\text{foc}}(X) = \cos \theta, \quad \text{where } \sin \theta = \frac{\|Z^T (I - XX^\dagger) Z\|_F}{\|XX^T - ZZ^T\|_F}.$$

This completes the proof. □

## Appendix B

In this section we provide a complete proof of Theorem 5, which includes all the intermediate calculations that was skipped in Section 5.3. We begin by proving a bound on $\sin \theta$.

**Lemma 13** (Same as Lemma 7). *Let $Z \neq 0$ and suppose that $\|XX^T - ZZ^T\|_F \leq \epsilon \|ZZ\|_F^2$. Then*

$$\sin^2 \theta = \frac{\|Z^T (I - XX^\dagger) Z\|_F^2}{\|XX^T - ZZ^T\|_F^2} \leq \frac{\epsilon}{2(\sigma_{\min}^2(Z)/\|ZZ^T\|_F) - \epsilon}.$$

*Proof.* The problem is homogeneous to scaling $X \leftarrow \alpha X$ and $Z \leftarrow \alpha Z$ for the same $\alpha$; Since $Z \neq 0$, we may rescale $X$ and $Z$ until $\|ZZ\|_F^2 = 1$. Additionally, we can assume that

$$X = \begin{bmatrix} X_1 \\ 0 \end{bmatrix} \qquad Z = \begin{bmatrix} Z_1 \\ Z_2 \end{bmatrix} \qquad \text{where } X_1, Z_1 \in \mathbb{R}^{r \times r}, Z_2 \in \mathbb{R}^{(n-r) \times r}$$

due to the rotational invariance of the problem. (Concretely, we compute the QR decomposition $QR = [X, Z]$ with $Q \in \mathbb{R}^{n \times 2r}$ noting that $X = QQ^T X$ and $Z = QQ^T Z$. We then make a change of basis $X \leftarrow Q^T X$ and $Z \leftarrow Q^T Z$). Then, observe that

$$\|Z^T (I - XX^\dagger) Z\|_F = \left\| \begin{bmatrix} Z_1 \\ Z_2 \end{bmatrix}^T \left( I - \begin{bmatrix} I & 0 \\ 0 & 0 \end{bmatrix} \right) \begin{bmatrix} Z_1 \\ Z_2 \end{bmatrix} \right\|_F = \|Z_2^T Z_2\|_F = \|Z_2 Z_2^T\|_F \quad (21)$$

and that $\|Z_2 Z_2^T\|_F^2 \leq \epsilon^2$ because

$$\|XX^T - ZZ^T\|_F^2 = \left\| \begin{bmatrix} Z_1 Z_1^T - X_1 X_1^T & Z_1 Z_2^T \\ Z_2 Z_{1in}^T & Z_2 Z_2^T \end{bmatrix} \right\|_F^2$$

$$= \|Z_1 Z_1^T - X_1 X_1^T\|_F^2 + 2\|Z_1 Z_2^T\|_F^2 + \|Z_2 Z_2^T\|_F^2 \leq \epsilon^2. \quad (22)$$

In order to derive a non-vacuous bound, we will need to lower-bound the term $\|Z_1 Z_2^T\|_F^2$ as follows

$$\|Z_1 Z_2^T\|_F^2 = \text{tr}(Z_1^T Z_1 Z_2^T Z_2) \geq \lambda_{\min}(Z_1^T Z_1) \text{tr}(Z_2^T Z_2) = \sigma_{\min}^2(Z_1) \|Z_2\|_F^2. \quad (23)$$

To lower-bound $\sigma_{\min}^2(Z_1)$, observe that

$$A + B \succeq \mu I \iff A \succeq \mu I - B \succeq (\mu - \|B\|_2) I,$$

and therefore

$$\sigma_{\min}^2(Z_1) = \lambda_{\min}(Z_1^T Z_1) \geq \lambda_{\min}(Z_1^T Z_1 + Z_2^T Z_2) - \lambda_{\max}(Z_2^T Z_2)$$

$$= \sigma_{\min}^2(Z) - \|Z_2 Z_2^T\|_2 \geq \sigma_{\min}^2(Z) - \|Z_2 Z_2^T\|_F$$

$$\geq \sigma_{\min}^2(Z) - \epsilon. \quad (24)$$

Finally, we substitute (21) and (22) and perform a sequence of reductions:

$$\frac{\|Z^T(I - XX^\dagger)Z\|_F^2}{\|XX^T - ZZ^T\|_F^2} = \frac{\|Z_2 Z_2^T\|_F^2}{\|Z_1 Z_1^T - X_1 X_1^T\|_F^2 + 2\|Z_1 Z_2^T\|_F^2 + \|Z_2 Z_2^T\|_F^2}$$

$$\overset{(a)}{\leq} \frac{\|Z_2 Z_2^T\|_F^2}{2\|Z_1 Z_2^T\|_F^2 + \|Z_2 Z_2^T\|_F^2} \overset{(b)}{\leq} \frac{\|Z_2 Z_2^T\|_F^2}{2\sigma_{\min}^2(Z_1)\|Z_2\|_F^2 + \|Z_2 Z_2^T\|_F^2}$$

$$\overset{(c)}{\leq} \frac{\|Z_2 Z_2^T\|_F \|Z_2\|_F^2}{2\sigma_{\min}^2(Z_1)\|Z_2\|_F^2 + \|Z_2 Z_2^T\|_F\|Z_2\|_F^2} = \frac{\|Z_2 Z_2^T\|_F}{2\sigma_{\min}^2(Z_1) + \|Z_2 Z_2^T\|_F}$$

$$\overset{(d)}{\leq} \frac{\epsilon}{2(\sigma_{\min}^2(Z) - \epsilon) + \epsilon} \leq \frac{\epsilon}{2\sigma_{\min}^2(Z) - \epsilon}.$$

Step (a) sets $X_1 = Z_1$ to minimize the denominator; step (b) bounds $\|Z_1 Z_2^T\|_F^2$ using (23); step (c) bounds $\|Z_2 Z_2^T\|_F \leq \|Z_2\|_F^2$ noting that a function like $x/(1 + x)$ is increasing with $x$; step (d) substitutes $\|Z_2 Z_2\|_F \leq \epsilon$ and $\sigma_{\min}^2(Z_1) \geq \sigma_{\min}^2(Z) - \epsilon$. $\qquad\square$