[Reviews · NeurIPS 2020]

Review 1

Summary and Contributions: This paper investigates matrix sensing. The problem is non-convex but it is known that if enough linear measurements are taken, then there are no spurious local minima making it amenable to optimization. In this work the authors characterize the number of measurements that are required to prevent any given point from becoming a spurious local minimum. Subsequently, they connect the quality of an initial point (in terms of how close it is to the true matrix) with the number of measurements needed to prevent the existence of spurious local minima within its neighborhood.

Strengths: I find this work very interesting in terms of understanding how the optimization landscapes gradually shapes into making a non-convex problem optimizable. The paper is theoretically sound and extends results that were previously known only for matrices with rank 1. I believe that understanding the landscape of non-convex optimization problems, as well as why and under what initial conditions we can efficiently solve them constitute important questions that are core and of interest to the NeurIPS community.

Weaknesses: There are practical questions that emerge from this work that remain unanswered, for example, how would you find such an initial point that is good, in order to reduce the number of measurements that you need? However, I think that the value of this work is mostly on providing a theoretical understanding for the evolution of the landscape and less on being a practical solution. In a similar note, I have a question regarding the optimization: we know that if we have enough measurements we will not have a spurious local minimum in a ball around the true solution. However, does this imply that gradient descent, will converge to the true minimum? There could be other spurious local minima outside of the ball and there is no guarantee as to where GD will take us, in principle it can take us to a spurious local minimum outside of the ball, right?

Correctness: The paper is quite technical, as far as I can tell the claims seem correct and believable, same for the high-level ideas. I didn’t verify the appendix proofs.

Clarity: Overall, the paper is well-written. I would say that the related work is in an awkward place, I’d prefer to see it closer to the introduction (especially given that it mentions the contributions of the current paper in a high-level while it follows Section 3 where the authors presented the contributions formally), or at the end. Now it feels that it cuts the paper in half.

Relation to Prior Work: The authors explain how their current work extends previous results on both local and global guarantees and how their techniques differ from the ones of Zhang et al. that appears to be one of the most related works.

Reproducibility: Yes

Additional Feedback: Overall, I enjoyed reading this paper and I think it makes interesting and important contributions to our understanding of non-convex problems. I would like to see it in the conference. Minor typos: - l123: \grad f_A = X) = 0 instead of \grad f_A^*(X) = 0 - l139: \delta* = 1/5 > 1 instead of for r > 1 - l174 that this is both and necessary and sufficient


Review 2

Summary and Contributions: In this work, the authors came up with a closed-form expression for the threshold on the number of samples needed to guarantee no spurious local minima for the nonconvex matrix recovery problem. It provides an important trade-off between the quality of the initial point and the sample size.

Strengths: The paper is well written. I briefly went over the proofs and they looked correct. They were carried out systematically. Moreover, the geometric intuition behind the closed-form formulation of the lower bound on the local threshold made the analysis even more clear. In addition, this work also simplifies and extends the proof of [23] to accommodate the case where the rank is greater than unity. Overall, I think this work is going to be an important addition in terms of understanding the significance of a good initialization in nonconvex setup.

Weaknesses: This is not exactly a weakness. But I think title of the paper is way too generic. It would be great if the authors could make it a bit more problem specific.

Correctness: I think the paper is very well written and well presented and I enjoyed reading it.

Clarity: See above.

Relation to Prior Work: The authors understand many of the underlying subtleties and know the background literature well, and are able to use very specific results from the literature.

Reproducibility: Yes

Additional Feedback: The authors have addressed my comments in their response.


Review 3

Summary and Contributions: This paper considers a low-rank matrix sensing problem from linear measurements and studies the sample complexity needed to guarantee no spurious local minimum in a given parameter region. The approach leverages the notion of RIP constant of the measurement matrix.

Strengths: The theorem in this paper provides a restricted isometry constant threshold for matrix sensing problem which can guarantee that the true solution is the unique local/global minimum point within a neighborhood of the true solution, where the neighborhood radius epsilon and the rank r of the input matrix can be arbitrary. Hence, this theoretical result is more flexible and general than the previous ones, and thus provide insights on the trade-off between sample complexity requirement and initialization quality in the matrix sensing problem.

Weaknesses: (1) More information could be added to the introduction of the matrix sensing problem. For example, are A_1…A_m known? Are they fixed or random? What are applications of matrix sensing and maybe what is its relationship to machine learning? Such information perhaps makes this topic clearer and more motivating, especially to the people who touch this topic for the first time. (2) The key idea of this paper is to propose to use the sample complexity for eliminating spurious critical points in a given region and the sample complexity for eliminating the spurious local min globally as a lower bound for the sample complexity for eliminating local min in a given region, see eq (7). This is a correct lower bound. But I did not see why it is a tight one as the author claimed from Fig1. Is it a numerical evaluation or a rigorous proof? (3)The paper focuses on low-rank matrix sensing problem, which has a relatively simple loss that leads to closed form solutions. Can the results of this paper have any implications on deep models? In deep learning, one usually applies Gaussian-type initialization and it seems that the training can always find a global min. (4) I want to point out that the notion \delta_soc(W) can eliminate spurious local min for any set W. But the sample complexity lower bound obtained in the paper only applies to W that is close to the ground truth. I suggest the author clarify this. Moreover, existing studies on matrix sensing and phase retrieval have established sample complexities for guaranteeing the condition eq(9) by proving local strong convexity-related conditions. In this perspective, how does the main result here strengthen the existing results? (5) The numerical example is an explanation of theorem 5 under the specific 1-rank case. I think it may be better to demonstrate the theorem via more experiments. For example, for a certain (X,Z), use Corollary 6 to calculate the threshold for m, the number of samples. Then, select a grid of values of m, some below the threshold and some above. For each value of m, conduct multiple optimizations using randomly generated A_1… A_m and initialization X, and see how many experiments converges to Z. Then, show the proportion of the experiments that converges to Z for each value of m. Can it reach 100% when m exceeds the threshold?

Correctness: Yes, I think the theoretical method and empirical example are both correct.

Clarity: Yes, I think the paper is well written.

Relation to Prior Work: Yes, I think it clearly discussed the contribution of this work from the previous works.

Reproducibility: Yes

Additional Feedback: (1) What is the value of C_0 in Corollary 6?


Review 4

Summary and Contributions: This paper studies the low-rank matrix sensing problem. It characterizes a lower bound on the RIP constant so an arbitrary point is not a spurious local minimum. In order to make the problem more tractable, the authors relaxed the condition so they found a lower bound on the constant where the arbitrary point is not even a stationary point. The main contribution of the paper would be that the RIP condition can be relaxed from the global guarantee by Bhojanapalli et al. if we have an initial guess closer to the solution. This leads to relaxed sample complexity by a constant factor, i.e., m > C(\eps) nr (m : # samples, n : dimension, r : rank, \eps : distance between the initial guess and the solution, C(0) is still non zero constant)

Strengths: 1. Understanding the landscape of low rank matrix problems has been of interest to this community. Many inspiring results have been published recently. 2. The key result establishes a relation between the basin of attraction and RIP condition, and the fact that the RIP condition can be relaxed when the initial guess is close to the global solution. This characterization wasn't much discussed in the existing work.

Weaknesses: 1. I wonder if this work should also discuss how harder it gets to obtain such a good initial point (within B_\eps). While we characterized that we only need relaxed RIP condition when the initial point is closer to the global optimum, but wouldn't we need a stronger RIP condition to get such a good initial point? I wonder if the motivation of this work is only establishing a theoretical relationship between the RIP condition and the basin of attraction, or also related to any practical applications where this results can be effective. 2. Considering the existing analysis in the literature, I think that the analytical result is a bit incremental. The implication of Theorem 4 looks similar to [1, Lemma 4.2], which could also be interpreted as a lower bound on the RIP constant for a matrix to be a stationary point. And it can also lead to a similar property to L47: "a point X is more likely to be a local minimum if ... is closer to orthogonal", though the lemma wasn't used for this purpose. I couldn't find an interesting analysis technique for NeurIPS publication. [1] Bhojanapalli et al, "Global Optimality of Local Search for Low Rank Matrix Recovery"

Correctness: The analysis made sense, thought I couldn't fully checked the proof line by line.

Clarity: This paper is clear and well written. I didn't get any concerns in its clarity when reading the paper.

Relation to Prior Work: It is fairly clear that how this work differs from existing works.

Reproducibility: Yes

Additional Feedback:

[Author Response · NeurIPS 2020]

We thank the area chair and the four reviewers for their careful reading and helpful comments. We will begin with some
general clarifications and then follow with specific response line by line.

**Our Contributions** For the non-convex problem of matrix sensing, we define a function $\delta_{\mathrm{soc}}(X)$ that gives a *precise*
threshold on the number of samples need to prevent $X$ from becoming a spurious local minima. Although $\delta_{\mathrm{soc}}$ is
difficult to compute exactly, we obtain a *closed-form*, sharp lower bound using convex optimization. As a result, we are
able to characterize the *tradeoff* between the quality of the initial point and the sample complexity.

**Comparison with previous results on local convergence:** Various previous works have shown that linear convergence
occurs around a small, fixed neighborhood of the global min (see Bhojanapalli et al., Tu et al., etc). The proof techniques
are similar: restricted local convexity holds when the sample size is sufficiently large. However, these proof techniques
are incapable of charactering how the optimization landscape changes as sample complexity increases. Our work paints
the full picture: the problem becomes more 'non-convex' (requiring more samples to eliminate spurious local min)
as we get further and further away from the global min. Once outside $\mathcal{B}_\varepsilon$, it becomes *necessary* to rely on the global
guarantees of Bhojanapalli et al. In contrast, previous work on local convergence only show convexity in a small
neighborhood, and tells us nothing about the landscape outside that small neighborhood.

**How to find an initial point:** As reviewer 1 points out, the main concern of our paper is understanding how the
landscape changes with sample complexity. Therefore, we chose to view the initial point as a part of the *problem*
*structure*. Nevertheless, there is a substantial body of previous work (e.g. Bhojopanalli et al., Tu et al., Candes et al.)
that separately studies the problem of finding a good initialization. One possible difficulty, as reviewer 4 notes, is that
some of these methods, such as spectral initialization, already require a large sample size. *But we emphasize that this*
*is not the only way to get an initial point.* For example, matrix sensing arises in the electric grid application under
the name "state estimation". Here, the ground truth corresponds to a physical quantity of interest. Domain-specific
heuristics that depend on physical and engineering intuition are able to deliver high quality initial points that are then
further refined via non-convex optimization.

**Response to reviewer 1:** We thank the reviewer for the nice summary of our paper. We will move the related works
section towards the end of the paper. Regarding the second question in section 3, we note that GD will always stay in
the $\varepsilon$-ball when the sample size is large (but still on the order of $O(nr)$). In this case the inner product between $\nabla f(X)$
and $\nabla \|XX^T - ZZ^T\|_F^2$ is always positive. When the sample size is smaller, we can rely on problem structures to
prevent the algorithm from leaving the neighborhood. For instance, with any descent algorithm, we are guaranteed to
stay in the region if we initialize within a smaller interior (See [23]).

**Response to reviewer 2:** We thank the reviewer for the positive feedback. We agree that the title of the paper is indeed
too general and we will change it to *How Many Samples is a Good Initial Point Worth in Low-Rank Matrix Recovery?*

**Response to reviewer 3:** We thank the reviewer for very detailed comments. (1) We agree that more motivation should
be provided for the matrix sensing problem. We have added a brief section in the intro that discusses the application
of matrix sensing in problems like quantum state tomography, metric learning, and electric grids. We also clarified
our assumptions: the measurement matrices $A_1, \ldots, A_n$ are fixed, and can be from *any* RIP ensemble. (2) Regarding
the tightness of our lower bound: the plot in figure 1 shows the rank-1 case, where the bound has been shown to
be tight for all $\varepsilon$ (See [23]). In the high-rank case, $\delta_{\mathrm{foc}}$ is very close to 1 when $\varepsilon$ is small, as indicated by Theorem
5. Since $\delta_{\mathrm{foc}} \leq \delta_{\mathrm{soc}} < 1$, the gap between $\delta_{\mathrm{foc}}$ and $\delta_{\mathrm{soc}}$ is small. When $\varepsilon$ becomes large, we switch to the *global*
lower bound $\delta_{\mathrm{soc}}(\mathbb{R}^{n \times r}) = 1/5$, which is again exactly tight. (3) Arguably, matrix sensing is one of the handful
non-convex problems that admits rigorous theoretical analysis, and our work provides deeper understanding of how
non-convexity can be overcome with more training samples. We believe this is an important step towards understanding
the relationship between sample complexity and the optimization landscape in deeper models. (4) Notice that when the
number of measurements is below the threshold defined by $\delta_{\mathrm{soc}}$, our results guarantee that there *exists* some choice
of the measurement ensemble $\mathcal{A}$ such that the problem will have a spurious local minima. However, sampling from
sub-Gaussians distributions in general does not find these adversarial cases. This is indeed a subtle point, and we have
added a brief discussion in the numerical results section.

**Response to reviewer 4**: We thank the reviewer for the helpful comments. For the concerns raised in section 3, please
refer to our discussion at the beginning. We emphasize that our main contribution is *not* improved RIP-conditions.
Rather, it is a new proof technique that establishes a *tradeoff* between sample complexity and the quality of the initial
point. This is something that previous methods based on local convexity are incapable of characterizing, since their
analysis depends on a *fixed* neighborhood. Note that lemma 4.2 in [1] only bounds the distance in the *subspace* spanned
by the column of $U$, and the error along the orthogonal direction can still be large. Therefore, this lemma can't actually
eliminate spurious critical points, even when $\delta$ is arbitrarily small. In contrast, our analysis finds the precise number of
samples to prevent *any* point from becoming a spurious critical point, allowing us to describe how the optimization
landscape 'evolves' as sample complexity increases.

[Meta-Review · NeurIPS 2020]

This paper tries to quantify the effectiveness of good initial solution in the context of matrix factorization/sensing problems. More concretely, they showed that with a good initialization the RIP condition required for matrix factorization/sensing can be relaxed, resulting in requiring a smaller number of samples. Overall the reviewers found the perspective interesting and the results are clean.